# Clinical Efficacy of Mesenchymal Stem Cells in Bone Regeneration in Oral Implantology. Systematic Review and Meta-Analysis

**DOI:** 10.3390/ijerph18030894

**Published:** 2021-01-21

**Authors:** Sonia Egido-Moreno, Joan Valls-Roca-Umbert, Juan Manuel Céspedes-Sánchez, José López-López, Eugenio Velasco-Ortega

**Affiliations:** 1Department of Odontoestomatology, Faculty of Medicine and Health Sciences, School of Dentistry, University of Barcelona—Campus Bellvitge, 08907 Barcelona, Spain; soniaegido@ub.edu (S.E.-M.); joan.valls@yahoo.es (J.V.-R.-U.); Juanmacesp88@hotmail.com (J.M.C.-S.); 2Department of Stomatology, Faculty of Dentistry, University of Seville, 41013 Seville, Spain; evelasco@us.es

**Keywords:** dental implants, regeneration, stem cells, bioengineering

## Abstract

In bone regeneration, obtaining a vital bone as similar as possible to native bone is sought. This review aimed to evaluate the efficacy of stem cells in maxillary bone regeneration for implant rehabilitation and to review the different techniques for obtaining and processing these cells. A systematic review and meta-analysis were performed using the Pubmed/Medline (NCBI), Cochrane, Scielo, and Scopus databases, without restriction on the publication date. The following Mesh terms were used, combined by the Boolean operator “AND”: “dental implants” AND “stem cells” AND “bioengineering”. Applying inclusion and exclusion criteria, five articles were obtained and three were added after manual search. The results from the meta-analysis (18 patients) did not provide significant differences despite the percentage of bone formed in the maxillary sinus, favoring the stem cell group, and the analysis of the percentage of residual Bio-Oss^®^ showed results favoring the control group. Stem cell regeneration usually shows positive vascular and viable bone formation. In conclusion, using mesenchymal stem cells in bone regeneration provides benefits in the quality of bone, similar or even superior to autologous bone, all this through a minimally invasive procedure.

## 1. Introduction

Nowadays, dental implant rehabilitation of edentulous patients is very widespread. Osteointegrated dental implants are increasingly used to replace missing teeth in a variety of situations, ranging from a missing single tooth to complete edentulism [1]. Osseointegration in clinical dentistry depends on an understanding of the healing and reparative capacities of hard and soft tissues [2]. Dental implants have several advantages over other therapies: high success rate and improvement in esthetics, chewing, eating, or speaking [3]. Nevertheless, bone availability for a successful treatment is not always ideal. To solve this situation, guided bone regeneration can provide us the necessary bone volume, which is a conditioning factor for success [4,5].

In the past, surgeons resorted to synthetic materials, derived from other species and allografts. These materials allow an increase in bone volume but lack osteogenic capacity, which results in longer healing periods and lower bone quality regeneration, which is less vascularized, and has worse biomechanical capacities [6]. The absence of osteogenic cell migration, such as osteoblasts and vascular precursor cells to the interior of the regeneration, can imply its failure [5,6,7].

In order to achieve a successful regeneration, we require three essential processes: (i) the presence of the appropriate conditions for the growth of osteogenic elements (osteoconduction); (ii) the stimulation of undifferentiated cells to form osteoblasts (osteoinduction); and (iii) the presence of osteoblasts that will form new bone (osteogenesis) [6]. At present, the gold standard for bone regeneration is the use of autologous bone, due to its excellent osteogenic capacity, shorter healing time, predictability, and being rejection free [5,8]. The classic way to obtain an autologous graft is by performing a second surgery, to recollect the bone block or particles, which in many cases are in a very limited quantity. This is a more invasive technique, with longer surgery time, worse postoperative pain, and possible aftermath complications [4,6,9]. The rate of intra- and postoperative complications (bleeding, pain, dysesthesia, infections) derived from the autologous bone graft procurement can be as high as 20% [6].

The use of stem cells obtained from bone marrow aspirate from the anterior or posterior iliac crest or even the breastbone can be presented as a complement or even a substitute for the use of autologous bone. This technique is less invasive, attaining the same advantages as the autologous bone graft and bypasses the morbidity associated with the autologous graft [7].

The cells attained from the bone marrow are pluripotential cells that can differentiate into multiple types of cells, such as osteoblasts, adipocytes, and chondrocytes, amongst others [7,10]. When they are used in combination with 3D calcified matrixes constituted by biomaterials within a defect that requires regeneration, they will form primary bone tissue that is highly vascularized [4]. This combination has been proven to form higher quality bone more quickly, in comparison with the use of only non-autologous biomaterial [6]. A viable option to obtain this 3D calcified matrix is the use of 3D bioprinters that, despite being in the early stages of development, may become the perfect alternative for bone defect regeneration [11]. Another viable technique to obtain bone morphogenetic proteins (BMPs) directly in the regeneration area is the use of hyaluronic acid hydrogel with ZrO_2_ as a vehicle to enhance bone formation [12,13]. Furthermore, the differentiation capacity of autologous stem cells and their capacity to stimulate the mobilization of other stem cells and progenitor endogenous cells towards the bone defect will increase the number of osteoblasts within the regeneration and, in addition to their immunomodulation capacity, they play an important role in the result of the regeneration [6].

The pluripotential capacity of stem cells, despite being a great benefit, is also its Achilles’ heel. Achieving differentiation into osteoprogenitor cells is one of the challenges that must be faced to achieve successful bone regeneration because the specific cellular characteristics will influence the regeneration results [6,7]. These cells must be previously centrifuged to concentrate them and cultured in a medium that favors their differentiation into the osteogenic cell line and then implanted in the receptor bed [6].

## 2. Materials and Methods

### 2.1. Search Strategy, Article Selection, and PICO Question

Bibliographic research was carried out in the Pubmed/Medline (NCBI), Cochrane, Scielo, Web of Science (WOS), Cumulative Index to Nursing and Allied Health Literature (CINAHL), and Scopus databases. The research was performed until October 31, 2020, without applying restriction on the publication date. Articles that contained information about the use of stem cells obtained from the bone marrow for bone regeneration in the jaws were searched. The following Mesh terms were used combined with the Boolean operator “AND”: “*dental implants*” AND “*stem cells*” AND “*bioengineering*”. With this search strategy, all titles and abstracts found were evaluated and the bibliography of selected studies and other reviews was revised to include other relevant studies in our review.

Articles were selected by two blinded reviewers (Valls-Roca-Umbert J., (JVR) and Egido-Moreno S., (SEM)). Firstly, they reviewed titles and abstracts (phase-1). If papers were considered eligible for inclusion, a full-text reading was blindly performed by the same reviewers (phase-2). In case of disagreements, a third and fourth reviewer (Céspedes-Sánchez J.M., (MCS) and Velasco-Ortega E. (EVO)) were involved to make the final decision and it was agreed upon with López-López J. (JLL).

In this work, we will focus on those studies that analyze the use of mesenchymal stem cells (MSCs) for the regeneration of maxillary bone defects. We set ourselves the objective of conducting a systematic review that answers the population, intervention, comparison, outcome (PICO) question: Does the use of stem cells in maxillary bone regeneration (I) improve the rate of bone formation, quality, and healing time (O), compared to regeneration that only uses autologous bone, biomaterial, or a combination of both (C), in patients without bone availability for implant placement (P)?

### 2.2. Inclusion/Exclusion Criteria

*Inclusion criteria*: Human studies; no language and time restrictions; clinical trials; articles that present information on the bone marrow aspiration site (iliac crest or sternum), method of obtaining it, processing, handling of the cells obtained, the type of regeneration, and results.

*Exclusion criteria*: Bibliographic reviews; meta-analysis; case report; case series; articles that did not deal with feedback; animal studies; studies with MSCs provided by pulp/adipose tissue.

### 2.3. Risk of Bias

The clinical trials included in the review were assessed using the Jadad scale [14] and with the Risk of Bias in Non-randomized Studies of Interventions (ROBINS-I) assessment tool [15] and the review itself was assessed using the Preferred Reporting Items for Systematic Reviews and Meta-Analyses (PRISMA) scale [16].

### 2.4. Statistical Analysis

The tool used for the statistical analysis was the OpenMetaAnalyst program (Center for Clinical Evidence Synthesis, Boston, MA, USA). Forest plots were produced to graphically represent the difference in results regarding the new bone formation and residual graft content in the grafts with a 95% confidence interval. *p* = 0.05 was used as the level of significance. Heterogeneity was assessed with the χ^2^ test and the I^2^ test.

## 3. Results

### 3.1. Search Strategy Results

Through our search strategy in Medline/Pubmed (National Center for Biotechnology Information, Bethesda, MD, USA), Scopus (Elsevier, Amsterdam, the Netherland), Scielo (Scientific Electronic Library Online, Madrid, Spain), WOS (Clarivate Analytics, London, UK), CINAHL (Universidad de Granada, Granada, Spain), and Cochrane databases (Cochrane Collaboration, London, UK), 382 articles were obtained. After eliminating the animal studies, 294 publications remained. After reading the titles and abstracts, 240 papers were discarded.

Of these, we eliminated 4 case series, 8 case reports, 16 reviews, and 18 studies that obtained stem cells from dental pulp or adipose tissue. Finally, after eliminating the duplicated articles, there were 6 articles [17,18,19,20,21,22] that met our inclusion criteria. Two additional articles resulting from the manual search were included for the interest of the review (Rickert D et al. [23] and Wildburger A et al. [24]) (Figure 1).

### 3.2. Risk of Bias

Table 1 represents the methodological quality of the selected studies using the JADAD scale [14], Table 2 represents the Risk of Bias in Non-randomized Studies of Interventions (ROBINS-I) assessment tool [15], and the evaluation using the PRISMA scale [16] met 22 items.

### 3.3. Demographic Results

The included population comprised 110 patients who underwent 128 regenerations. The mean age of the patients was 51.82 ± 9 years; however, Shatayeh et al. [22] did not provide this information. Regarding sex, three [20,23,24] of the selected articles did not provide us data in this regard, in the remaining studies, the population studied was composed of 27 (36.5%) men and 47 (63.5%) women.

### 3.4. Regeneration Results

Of the 128 regenerations presented, 89 were performed using MSCs mixed with biomaterials (beta tricalcium phosphate (β-TCP), BioOss^®^ (Geistlich Bio-Oss^®^, Inibsa Dental, Barcelona, Spain)), allogeneic bank bone, or biphasic calcium phosphate (BCP) as a scaffold for the grafted cells and 39 were done only with regeneration techniques that did not involve the use of stem cells as a control.

The mean follow-up was 18.25 months (4 months to 5 years).

Regarding the areas where the regenerations were performed, five studies analyze maxillary sinus elevations [17,18,20,22,23,24], two articles study bone ridge augmentation regenerations in the maxilla [21] and mandible [19], and, finally, one study analyzes regeneration in patients with cleft palate and trauma [21].

The areas where the medullary aspirate was obtained were the posterosuperior part of the iliac crest [17,18,19,20,21,22,23] and the pelvic bone, 2 cm posterocaudal to the iliac crest [24].

Depending on the study, the amount of marrow aspirate ranged from 5 to 70 mL [17,18,19,20,21,23,24]; this volume depends on the location of the medullary aspirate as well as the collection method (Table 3).

#### 3.4.1. Stem Cell Processing, Culture/Centrifugation Method, Cells Obtained

All the articles use methods of isolation, expansion, and characterization of the stem cells collected from the bone marrow. Different methods were used for the processing of stem cells, including the Ficoll-Paque™ system by gradient differentiation by centrifugation [17,20,21], the bone marrow aspirate concentrate (BMAC) system [20,23,24], isolation of the cells present at an interface density of 1.073 g/mL [18], another by density gradient centrifuge at 750 g [22], and one last study differentiated the cells only by the medium in which they were cultured [19].

Afterward, these cells were incubated in a conductive medium for the differentiation of the cell line that favored the formation of osteoprogenitor cells: Dulbecco’s modified Eagle’s medium [18,22], Iscove’s modified Dulbecco’s medium [17,21], and modified minimal essential medium alpha [19]. In the study by Kaigler et al. [17], they did not create a culture medium, but after eliminating the non-pluripotent cells by filtering through a membrane with a porosity of 80μm, they already achieved the fixation of the MSCs on the biomaterial scaffold by applying two glycoproteins (fibronectin and laminin) on it. The studies [20,23,24] that include the BMAC process in their protocol did not use a culture medium for the proliferation of the cell lineage, since it is a method that manages to isolate the MSCs from the medullary aspirate by employing a specific centrifugation protocol.

The type of cells obtained after the process selected by each study is expressed in different ways: utilizing colony-forming units (CFUs) [20,24]; biomarkers such as CD14, CD44, CD90, CD73, or CD105, among others [17,19,21,22,23,24]; with the expression of alkaline phosphatase, collagen, or calcification [18,20]; or even cells per mL [17,18,19,21,24]. Kaigler et al. [17] confirmed the presence of mononuclear cells with an electron microscope, eosin staining, and immunostaining for SH-2, SH-3, and SH-4 antigens. The positive relationship between the number of CD90 + MSCs and bone quality is of notable relevance; the higher the percentage of these cells, the better the bone quality obtained [17] (Table 4).

#### 3.4.2. Biomaterial Characteristics

The proportion of MSCs in each graft varies depending on the amount of biomaterial required to regenerate the defect, in addition to factors such as the patient, the processing method, and the concentration of mononuclear cells [17,20,21].

The concentrations of the different biomaterials vary according to the study: Three studies [18,23,24] used BioOss^®^; 2 g of BioOss^®^ (0.25–1 mm, particle size) with 3 mL of mononuclear cells (40 × 10^6^ cells / mL specified in the study [24]) with 1 mL of human thrombin [23,24], and the same combination but without specifying quantities [18]. In two studies [17,21], β-TCP from Cerasorb^®^ was used; β-TCP in a 1:1 volume ratio with mononuclear cells, β-TCP in a 1:1 volume ratio with mononuclear cells, a total of approximately 15–80 × 10^6^ cells per graft (5–10 × 10^6^ cells / mL) [17]; and 2–5 mL of β-TCP with 15–44 × 10^6^ cells/mL of mononuclear cells [21]. The study by Ueda et al. [18] included a combination of 3.5 mL platelet-rich plasma (PRP) with 1 × 10^7^ mononuclear cells/mL mixed with 500μL of calcium chloride and thrombin, for a total of 1.5–5.8 gr per graft. Gjerde et al. [19] combined 5 mL of BCP (MBCP^®^) consisting of 20% hydroxyapatite and 80% β-TCP (in 0.5–1 mm granules) with 100 × 10^6^ mononuclear cells (20 × 10^6^ cells/mL). In the study by Sauerbier et al. [20], a combination of BioOss^®^ (0.25–1 mm) with thrombin-enriched mononuclear cells was used, without specifying the number of cells per mL. Finally, Shayetesh et al. [22] used cubes composed of β-TCP/hydroxyapatite, as a scaffold, mixed with 0.2 mL (5 × 10^5^ cells) of MSCs.

The authors stated that the key factors to take into account when selecting the biomaterial are particle size, porosity, reabsorption time, and the amplitude of the defect to be regenerated [21,24]. The size of the particle determines the regeneration, as the smaller the size, the greater the compactness of the particles, and the formation of new bone is prevented. Likewise, porosities lower than 100μm do not allow cell or capillary invasion [24]. As in the study by Ueda et al. [18], the regeneration of large defects does not achieve favorable results since the use of β-TCP is reabsorbed by 90% in 3 months, due to not being able to maintain graft stability long enough for neo-ossification.

The incubation time and temperature to which the cell concentrate is subjected, together with the selected biomaterial, are essential for the success of regeneration. For example, with β-TCP, to achieve greater survival and adhesion of the cells to the biomaterial, both must be incubated in contact with each other for 30 min at room temperature or 4 °C, and if the time increases to 1 h, the temperature must be 4 °C to ensure cell viability [20]. The ideal amount of mononuclear cells per unit volume of β-TCP is between 15–44 × 10^6^ cells/mL [21].

#### 3.4.3. Regeneration Outcomes

Concerning the results obtained after the different regenerations with stem cells, all the articles analyzed [17,18,19,20,22,23,24] coincide on a trend towards positive bone formation, except for one [21]. In this study [21], good results were not obtained when regenerating defects caused by trauma or in cases of cleft palate, due to the poor stability of the particulate graft in the defect compared to autologous bone block grafts. They manage to regenerate only 0.5–2 mm in patients with cleft palate and 0.5–5 mm in trauma patients, an amount they consider insufficient [21]. The rest of the regenerations show a good gain in volume, bone width, and bone quality for the authors to describe the regenerations as a success [17,18,19,20,22,23,24]. In an article where bilateral sinus lift is compared with BioOss^®^ and stem cells as opposed to using only BioOss^®^, it is found that the gains in the volume are similar, but not in the quality of the bone formed, with the one enriched with stem cells being statistically significantly more favorable [17]. Likewise, there is a statistically significant difference in bone quality between sinus lifts with BioOss^®^ + stem cells, compared to BioOss^®^ + 70/30 autologous bone [23]. In contrast to these results, despite achieving a successful regeneration, one article shows no significant difference favoring the use of stem cells to achieve better bone quality and a shorter bone formation time [24].

Regarding new bone formation, studied by four articles [20,22,23,24] that comprise a total of 60 regenerations (42 in test groups and 18 in control groups), the new bone formation was 22, 94 and 13.8% for each article, respectively (Figure 2).

To assess the type of bone formed, seven studies [17,19,20,22,23,24] took a biopsy for histological analysis between 3 and 8 months after regeneration. The histological description agrees between the different studies: the presence of mature lamellar bone tissue with bone trabeculae containing osteoblasts, richly irrigated by vascular structures, with integrated biomaterial particles that are in different states of reabsorption by osteoclasts, with bone tissue on its surface and between the pores [17,19,20,24]. Newly formed bone tissue is of a notably higher quality when it is supplemented with stem cells, compared to defects regenerated only with biomaterial, because it presents greater vascularization, a greater quantity of mineralized tissue, and greater cellularity, especially in large regenerations [17,24]. Bone formation over time is more predictable and faster, and in 6 months a greater volume of bone is achieved compared to xenografts in those regenerations that use stem cells, in addition to observing a progressive decrease in the volume of biomaterial particles [23,24].

### 3.5. Meta-Analysis of the Results

Only two articles [23,24], that make up a total of 25 patients and 36 regenerations, allow us to analyze the percentage of bone formation in the maxillary sinus. The result favors the stem cell group, although it is not statistically significant (weighted difference of means (WMD): 3.187; 95% CI: −2.714 to 9.088, *p* = 0.296 and I^2^ heterogeneity: 39.03%, *p* = 0.2) (Figure 3).

Regarding the analysis of the percentage of residual BioOss^®^ at 3–4 months, this is analyzed by two studies [23,24] composed of 25 patients and 36 regenerations. The result shows us a difference in favor of the control group (WMD: 2.402; 95% CI: −6.052 to 10.857, *p* = 0.578 and heterogenicity: I^2^ = 73.17%, *p* = 0.054) (Figure 4).

## 4. Discussion

This review aimed to evaluate the efficacy of the use of stem cells in maxillary bone regenerations for subsequent implant rehabilitation, as well as to review the different techniques for obtaining and processing these cells.

Stem cell harvesting is done by bone marrow aspirate. This technique presents a low rate of complications (0.05%) and morbidity for the patient [25], which gives it a clear advantage over techniques for obtaining autologous bone [6,20], which frequently involve minor complications (9–39%) and occasionally severe complications (0.76–25%) [26]. The volume of the medullary aspirate is determined by the size of the defect to be regenerated. There is no optimal aspirate volume, but a volume between 2 and 4 mL is considered to provide >85% of aspirated stem cells in a given location, and will therefore be sufficient for most regenerations [8,27].

The great limitation of the use of MSCs is the need to cultivate them before being grafted into the recipient bed. As they have pluripotential capacity, the cells must differentiate mainly into bone-forming cells and no other tissues, achieving the mononuclear fraction of the medullary aspirate [19,28]. Different culture media and centrifuge forms have been developed to obtain osteogenic cells, and one of the most widespread methods is the FICOLL (Bone marrow-derived mononuclear cell isolation by synthetic polysaccharides) (centrifuge gradient of 1073 g/mL) [20,29]. However, it is currently being displaced by the BMAC method, a faster, cheaper, and more efficient way to obtain osteoprogenitor cells from the bone marrow aspirate [30]. The expression of alkaline phosphatase is an indicator of cell differentiation in osteoprogenitor cells (alkaline phosphatase is necessary for mineralization and regulates the function of osteoblasts, therefore, it is present when they differentiate into osteoprogenitor cells) [20,27,29], as well as the expression of biomarkers CD105, CD73, and CD90 [28]. The positive expression of biomarkers CD105, CD73, and CD90 are involved in the osteogenic process by promoting bone formation [31].

To a lesser extent, the need for a scaffold material, generally β-TCP or BioOss^®^, for the stabilization of stem cells in a defect, is another of the limiting factors for the use of this technique, as seen in the study by Bajestan et al. [21].

In our study, we evaluated new bone formation or the residual graft through a meta-analysis. No significant differences were found with the use of stem cells compared to other bone grafts [23,24].

Despite this, the use of stem cells is a favorable procedure that improves the rate of bone formation (Figure 3). The results are comparable to the use of autologous bone or are even improved, as we can see in the study by Rickert et al. [23], who found that in a period of 3–4 months, a higher percentage of bone is generated with a mixture of BioOss^®^ with stem cells than a mixture with autologous bone.

Regarding bone quality, one of the factors by which it is defined is by the amount of residual biomaterial. Finding a large amount of residual biomaterial leads us to find a lower quality of overall bone since there will be a subsequent lower quantity of neo-formed native vital bone. In our meta-analysis, we see that at 3–4 months, the BioOss^®^ content is lower in the control group. However, according to the results of Wildburger et al. [24], at 6 months, significantly less residual BioOss^®^ is found in the stem cell group. Another factor that determines bone quality is the bone volume fraction (BVF) and, as shown in the study by Kaigler et al., the stem cell group presents a statistically significantly higher BVF than the control group [17].

The use of MSCs competes with autologous grafting as the technique of choice, and some authors, such as Soltan et al. [10], even define it as the “platinum standard” for bone regeneration, presenting promising results compared to autologous bone grafting.

The study of the combination of stem cells with autologous bone could prove to be the true gold standard of bone regeneration, providing an essential scaffolding element for the use of stem cells, which in turn provides ideal properties for bone formation [5,6]. Therefore, more clinical trials are required to protocolize the use of stem cells in guided bone regeneration in maxillary defects.

Cell therapy is a continually growing field. The results obtained reveal the potential that stem cells provide to improve multiple treatment needs, such as bone regeneration for subsequent implant placement. Despite the benefits, the cost-effectiveness must also be taken into account, since currently the cost of this type of approach is very high compared to other techniques. For this reason, stem cell treatments are currently reserved for rare cases with low prevalence; therefore, it is crucial to develop new research to optimize procedures so that stem cell treatments can be used routinely [32].

This review has several limitations. First, the diversity in cell manipulation to obtain osteoprogenitor cells makes the results not comparable. Furthermore, we found that the studies dealt with regenerations of different types, with different degrees of complexity, on different bone structures, and there was a lack of homogenization between the studies in terms of the analysis of graft success; therefore, the results obtained cannot be standardized. Due to the heterogeneity of the results, we were only able to analyze two studies. Regarding the methodology used, we see that the methodological quality of some articles is low (Table 1) and some articles [18,19,20] lack a control group, so the comparison of their results with other methods of regeneration was not possible.

## 5. Conclusions

In conclusion, despite the limitations of the present study, bone regeneration through the application of stem cells in surgery and oral implantology show favorable results, although they are not always statistically significant. This is an innovative technique with a small number and low quality of trials, a lack of controls, and short follow-up periods. Aside from these limitations, this technique seems capable of providing us with an alternative to autologous bone grafts due to a low rate of morbidity and complications, and a reduction of the healing time and the capacity to form high-quality bone. More randomized clinical trials establishing homogeneous treatment protocols, with larger samples and longer follow-up periods, could lead us to unveil the real potential of MSCs to develop a minimally invasive treatment that favors high-quality bone tissue regeneration.

## Figures and Tables

**Figure 1 ijerph-18-00894-f001:**
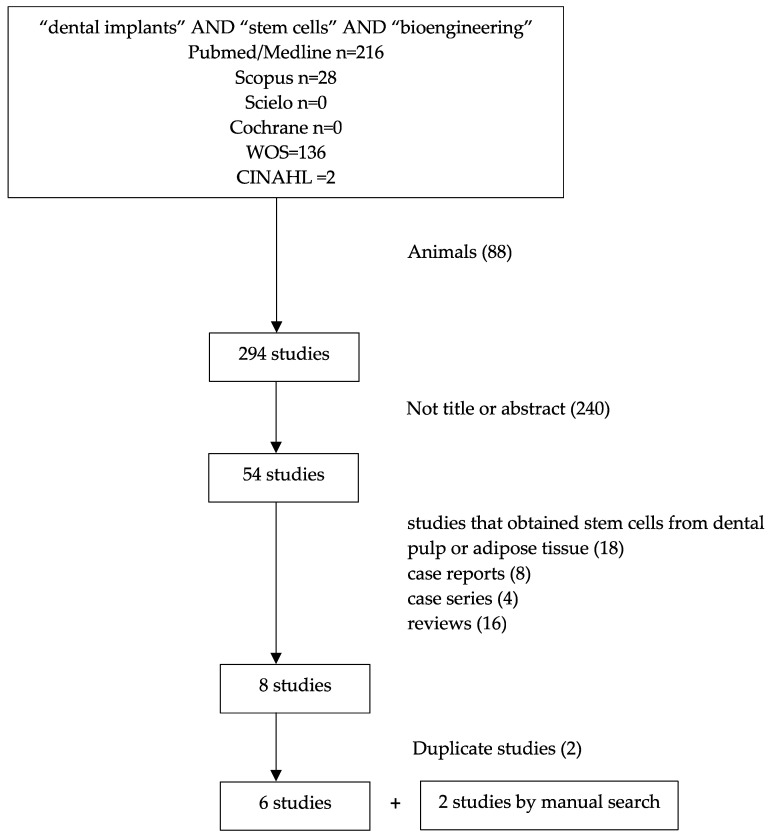
Flow chart illustrating the search strategy in the Pubmed/Medline database.

**Figure 2 ijerph-18-00894-f002:**
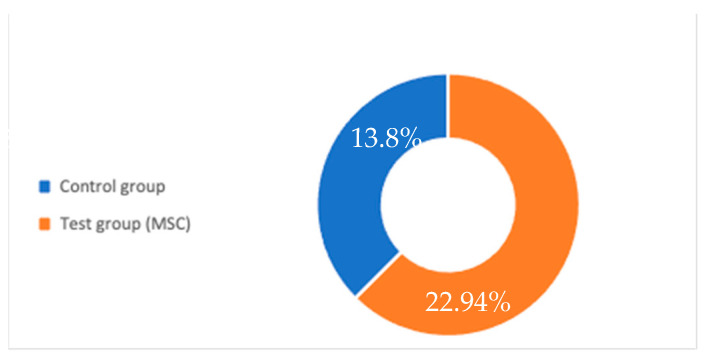
Comparative graphic between the control group and test group regarding new bone formation. MSC: Mesenchymal stem cell.

**Figure 3 ijerph-18-00894-f003:**
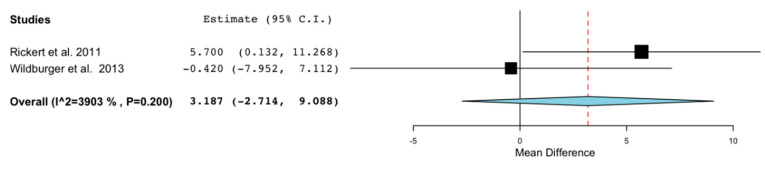
Forest plot of new bone formation between control vs. grafts with application of stem cells in maxillary sinus regenerations.

**Figure 4 ijerph-18-00894-f004:**
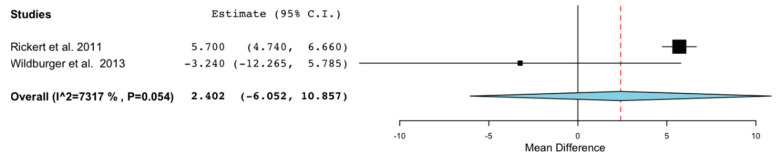
Forest plot of the residual content of biomaterial between control grafts vs. grafts with application of stem cells in maxillary sinus regenerations.

**Table 1 ijerph-18-00894-t001:** Quality evaluation of articles according to the Jadad scale [14].

	Randomized *	Double Blind *	Withdrawals and Dropouts *	Appropriate Randomization **	Appropriate Double Blind? **	Punctuation
Ueda et al. [18]	0	0	0	0	0	0
Shayesteh et al. [22]	0	0	1	0	0	1
Sauerbier et al. [20]	0	0	1	0	0	1
Rickert et al. [23]	1	2	1	1	0	3
Wildburger et al. [24]	1	0	0	1	0	2
Kaigler et al. [17]	1	0	1	1	0	3
Bajestan et al. [21]	1	0	1	1	0	3
Gjerde et al. [19]	0	0	1	0	0	1

* Yes: 1 point/No: 0 points; ** Yes: 1 point/No: 0 points/Inappropriate method: −1.

**Table 2 ijerph-18-00894-t002:** Quality evaluation of articles according to Risk of Bias in Non-randomized Studies of Interventions (ROBINS-I) assessment tool [15].

Ueda et al. [18]	Shayesteh et al. [22]	Sauerbier et al. [20]	Rickert et al. [23]	Wildburger et al. [24]	Kaigler et al. [17]	Bajestan et al. [21]	Gjerde et al. [19]	
								Random sequence generation (selection bias)
								Allocation concealment (selection bias)
								Blinding of participants and personnel (performance bias)
								Blinding of outcome assessment (detection bias)
								Incomplete outcome data (attrition bias)
								Selective reporting (reporting bias)
								Other bias

**Table 3 ijerph-18-00894-t003:** Characteristics of the regenerations.

	*n*	Age	Type of Regeneration	Place of Aspirate	Material	Average Gain (mm/cm^3^)	% Neo-formed Bone	Follow up
	sex			Quantity				
Ueda et al. 2008 [18]	14	54.6 (44–74)	Sinus lift (8)	Iliac crest	Stem cells + trombine + PRP + calcium chloride	Sinus lift: 8.7 mm	-	2–5 years
	W:12 M:2		Onlay graft (10)	10 mL		Onlay graft: 5 mm		
Shayesteh et al. 2008 [24]	6	-	Sinus lift	Iliac crest	Stem cells + β-TCP/ hydroxiapatite	8.58 mm	41.34%	12 months
	W:3 M:3			30 mL				
Sauerbier et al. 2010 [20]	18 C: 6 (FICOLL) T: 12 (BMAC)	C: 59.5 (50–69) T: 55 (47–68)	Sinus lift	Iliac crest	BioOss^®^ + stem cells + trombine	-	C: 15.5%	4 months
	-			60 mL			T: 19.9%	
Rickert et al. 2011 [23]	11 (C:11 T:11)	60.8 ± 5.9 (48–69)	Sinus lift (split mouth)	Iliac crest	C: BioOss^®^ + autologous bone T: BioOss + trombine + stem cells	-	C: 12 ± 6.6%	4 months
	-			-			T: 17.7 ± 7.3%	
Wildburger et al. 2014 [24]	14 (C: 7 T:7)	58 (47–72)	Sinus lift (split mouth)	Ilium	C: BioOss^®^ T: BioOss^®^ + stem cells	-	C: 13.9%	6 months
	-			-			T: 13.5%	6 months
Kaigler et al. 2015 [17]	26 (C:13 T:13)	C:49.1 (26–65) T: 53 (27–66)	Sinus lift	Iliac crest	C: β-TCP T: β-TCP + stem cells	C: 12.8 ± 2.8 mm C: 2.1 ± 0.9 cm^3^		1 year
	W:20 M:6			50–70 mL		T: 12.2 ± 3.3 mm T: 1.8 ± 1 cm^3^		
Bajestan et al. 2017 [21]	17 C:8 T:9	C:31 (19–54) T:27 (18–42)	C: Autologous block T: Onlay graft	Posterior part of iliac crest	C: Autologous block + allogenic particulate bone T:β-TCP + stem cells	C: 3.3 ± 1.4	-	6 months
	C: W:3 M:5 T: W:2 M:7			30–50 mL		T:1.5 ± 1.5		
Gjerde et al. 2018 [19]	11	65 (52–75)	Onlay graft	Iliac crest	BCP + stem cells	7.3 mm	-	3 years

W: Women; M: Men; PRP: Platelet-rich plasma; C: Control; T: Test; β-TCP: Beta tricalcium phosphate; BCP: Biphasic calcium phosphate; BMAC: Bone marrow aspirate concentrate; FICOLL: Bone marrow-derived mononuclear cell isolation by synthetic polysaccharides.

**Table 4 ijerph-18-00894-t004:** Characteristics of cell differentiations.

Study	CFU	Cells/mL	Biomarkers	Type of Cells
Ueda et al. 2008 [18]	-	1 × 10^7^	Alkaline phosphatase +	-
Shayesteh et al. 2008 [22]	-	5 × 10^5^	CD14	-
Sauerbier et al. 2010 [20]	28.7 ± 14.1 CFU/mL MNCs (BMAC)	-	Alkaline phosphatase +	Adipocytes, chondrocytes, osteoblasts, type I collagen
	25 ± 11.4 CFU/mL MNCs (FICOLL)	-	Alkaline phosphatase +	Adipocytes, chondrocytes, osteoblasts, type I collagen
Rickert et al. 2011 [23]	-	-	CD44 y CD73	MSCs differentiated into adipocytes, chondrocytes, and osteoblasts
Wildburger et al. 2014 [24]	4.43 CFU/mil MNCs	40 × 10^6^	CD90, CD73, CD105	-
Kaigler et al. 2015 [17]	-	5.15 × 10^6^	CD90, CD14	MSCs, monocytes/macrophages, mononuclear cells
Bajestan et al. 2017 [21]	-	14–44 × 10^6^	CD90, CD14	MSCs, monocytes/macrophages, mononuclear cells
Gjerde et al. 2018 [19]	-	20 × 10^6^	CD90, CD73, CD105, CD49D, CD19, CD34, CD45	-

CFU: Colony-forming unit; MNCs: Mononuclear cells; MSCs: Mesenchymal stem cells; BMAC: Bone marrow aspirate concentrate; FICOLL: Bone marrow-derived mononuclear cell isolation by synthetic polysaccharides.

## Data Availability

We did not upload statistical data but they are available to any author who requests them.

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
