# Peer review of "Clinical Efficacy of Mesenchymal Stem Cells in Bone Regeneration in Oral Implantology. Systematic Review and Meta-Analysis"

_ijerph, 2021, doi:10.3390/ijerph18030894_

Round 1
Reviewer 1 Report
Introduction provides sufficient details and it is fit for the topic. I may suggest adding a novel application of stem cells for future application that today appears promising that is 3D bioprinting. You may introduce this topic saying that up to now it is not suitable for extensive human application because of several inner limitation but it may be the future of the bone tissue regeneration (i.e. doi: 10.3390/ijms21197012). Methods appear adequately described. However I had some issue replicating your search in pubmed/medline. The string dental implants AND stem cells AND bioengineering retrieved only retrieved only 216 results in Pubmed. Could you please specify in more details the exact number of papers extracted from each database and then specify all the excluded papers and the reseon behind each exclusion (Pubmed / Medline (NCBI), Cochrane, Scielo and Scopus)? Results and Discussion appear well written and meaningful.
English style and grammar need minor revision
Author Response
Introduction provides sufficient details and it is fit for the topic. I may suggest adding a novel application of stem cells for future application that today appears promising that is 3D bioprinting. You may introduce this topic saying that up to now it is not suitable for extensive human application because of several inner limitation but it may be the future of the bone tissue regeneration (i.e. doi: 10.3390/ijms21197012). Methods appear adequately described. However I had some issue replicating your search in pubmed/medline. The string dental implants AND stem cells AND bioengineering retrieved only retrieved only 216 results in Pubmed. Could you please specify in more details the exact number of papers extracted from each database and then specify all the excluded papers and the reseon behind each exclusion (Pubmed / Medline (NCBI), Cochrane, Scielo and Scopus)? Results and Discussion appear well written and meaningful.
Thank you for your contributions.
-The authors reply:
# I may suggest adding a novel application of stem cells for future application that today appears promising that is 3D bioprinting. You may introduce this topic saying that up to now it is not suitable for extensive human application because of several inner limitation but it may be the future of the bone tissue regeneration (i.e. doi: 10.3390/ijms21197012).
It is a very interesting article; we have added some information in our manuscript. We have added the reference on page 2, we have included the text:
Lines 64-66: A viable option to obtain this 3D calcified matrix is the use of 3D bioprinters, that despite being in early stages of development may become the perfect alternative for bone defect regeneration.
Reference added:
-Genova T, Roato I, Carossa M, Motta C, Cavagnetto D, Mussano F. Advances on Bone Substitutes through 3D Bioprinting. Int J Mol Sci. 2020;21:7012.
# However I had some issue replicating your search in pubmed/medline. The string dental implants AND stem cells AND bioengineering retrieved only retrieved only 216 results in Pubmed. Could you please specify in more details the exact number of papers extracted from each database and then specify all the excluded papers and the reason behind each exclusion (Pubmed / Medline (NCBI), Cochrane, Scielo and Scopus)?
First of all, we want to apologize because que have made a mistake when uploading the data. We have corrected and detailed our search strategy in all data bases.
You could see the new flow chart in the document.
English style and grammar need minor revision
It has been read by a native colleague to revise the English

Reviewer 2 Report
The manuscript “Clinical application of mesenchymal stem cells in bone regeneration in oral implantology. Systematic review and meta-analysis” addresses an interesting topic and describes clearly the experimental methods and results.
Moreover, the paper also uses a good English. Therefore, it may be recommended for publication after minor revision:
Since the paper is an overview, in some points we could think that the deepness and references are too short. Nevertheless, the paper is well written and shows clearly understanding of the aim.
This reviewer thinks that the addition of some references could enhance the paper.
In introduction part, such as:
- CATAURO, Michelina, et al. Structure, drug absorption, bioactive and antibacterial properties of sol-gel SiO2/ZrO2 materials. Ceramics International, 2020;
- Materials, 2020, 13.2: 394.
Author Response
The manuscript “Clinical application of mesenchymal stem cells in bone regeneration in oral implantology. Systematic review and meta-analysis” address an interesting topic and describes clearly the experimental methods and results.
Moreover, the paper also uses a good English. Therefore, it may be recommended for publication after minor revision:
Since the paper is an overview, in some points we could think that the deepness and references are too short. Nevertheless, the paper is well written and shows clearly understanding of the aim.
This reviewer thinks that the addition of some references could enhance the paper.
In introduction part, such as:
- CATAURO, Michelina, et al. Structure, drug absorption, bioactive and antibacterial properties of sol-gel SiO2/ZrO2 materials. Ceramics International, 2020; Materials, 2020, 13.2: 394.
-The authors reply:
Thank you for your suggestions. We have added the study that you have recommended and some others to deepen more into the topic and give a broader vision of the subject. The added text and reference:
Another viable technique to obtain BMPs directly to the regeneration area is the use of hyaluronic acid hydrogel with ZrO2 as a vehicle to enhance the bone formation. [12, 13].
Bae MS, Kim JE, Lee JB, Heo DN, Yang DH, Kim JH, Kwon KR, Bang JB, Bae H, Kwon IK. ZrO2 surface chemically coated with hyaluronic acid hydrogel loading GDF-5 for osteogenesis in dentistry. Carbohydr Polym. 2013 Jan 30;92(1):167-75.

Reviewer 3 Report
This manuscript is a systematic review with meta-analysis about the effectiveness of the stem cells in bone regeneration to improve the quality of bone formation or accelerate the regeneration process. I consider that this review must be improved to be consider as potential publication in IJERPH:
ABSTRACT:
- Why the aim is not included?
- The results are poor. Why the meta-analysis findings are not included?
INTRODUCTION
- Poor references.
- Why implant therapy has not been explained?
- Why authors included the PICO in introduction? This must be included in methos section.
- I consider that is not necessary to include the null and alternative hypothesis. This is not an original study.
METHODS
- This review has not been registererd in PROSPERO or similar.
- The review's phases were made by peer?
- Insufficient databases. WOS and CINAHL Complete not included.
- Why authors don't have searched in grey literature?
- The search strategy proposed is very poor and non-specific.
- Please, improve the inclusion and exclusion criteria. Remember, exclusion criteria are not the opposite to inclusion criteria.
- I consider that include only studies published in english or spanish is very worring. It is a methodological mistake.
- To assess the quality with the JADAD scale is not sufficient. PRISMA review is not a quality assessment, unless a formal considerations to report the findings.
- Please, add more information about the statistical analysis. Publication bias risk? WMD?? Have you considered the precission level?
RESULTS:
- The studies included in the review analysis must not been included before the results. Authors use the studies included to write the introduction.
- Why in the PRISMA flow chart appear six studies and in the text only five?
- Report of the results improvable.
- What provided a meta-analysis with two studies? And with only one study?
- I would rethink a new way of writing the results, as well as assess the need for meta-analysis. A review with 5-6 + 2 studies is low in scope, but if done well it can be very interesting.
DISCUSSION
- I consider that the review is poor. the authors do not relate the results.
- Please, increase the references.
- I consider that this review have more limitations that provide it by the authors. Please, think it.
REFERENCES.
- Only 19 references in a review???
Author Response
Comments and Suggestions for Authors
This manuscript is a systematic review with meta-analysis about the effectiveness of the stem cells in bone regeneration to improve the quality of bone formation or accelerate the regeneration process. I consider that this review must be improved to be consider as potential publication in IJERPH:
Thank you very much for your comments.
-The authors reply:
# ABSTRACT:
Why the aim is not included? We have included the aim in the abstract (line 15-17)
The aim of this review was to evaluate the efficacy of stem cells in maxillary bone regenerations for implant rehabilitation and to review the different techniques for obtaining and processing these cells.
The results are poor. Why the meta-analysis findings are not included? To improve the abstract, we have included the findings of the meta-analysis in the abstract (line 21-24)
The results from the meta-analysis (18 patients) did not provide us with significant differences despite the percentage of bone formed in the maxillary sinus favoring the stem cell group and the analysis of the percentage of residual Bio-Oss® showed results favoring the control group.
# INTRODUCTION
Poor references: We have incorporated more references.
-Genova T, Roato I, Carossa M, Motta C, Cavagnetto D, Mussano F. Advances on Bone Substitutes through 3D Bioprinting. Int J Mol Sci. 2020 Sep 23;21(19):7012.
-Bae MS, Kim JE, Lee JB, Heo DN, Yang DH, Kim JH, Kwon KR, Bang JB, Bae H, Kwon IK. ZrO2 surface chemically coated with hyaluronic acid hydrogel loading GDF-5 for osteogenesis in dentistry. Carbohydr Polym. 2013;92:167-75.
-Catauro M, Barrino F, Dal Poggetto G, Milazzo M, Blanco I, Ciprioti SV. Structure, drug absorption, bioactive and antibacterial properties of sol-gel SiO2/ZrO2 materials. Ceramics International, 2020; Materials, 2020, 13.2: 394.
- Soria-Juan B, Escacena N, Capilla-González V, Aguilera Y, Llanos L, Tejedo JR, et al.;Collaborative Working Group “Noma Project Team”. Cost-Effective, Safe, and Personalized Cell Therapy for Critical Limb Ischemia in Type 2 Diabetes Mellitus. Front Immunol. 2019 Jun 4;10:1151.
-Shayesteh YS, Khojasteh A, Soleimani M, Alikhasi M, Khoshzaban A, Ahmadbeigi N. Sinus augmentation using human mesenchymal stem cells loaded into a beta-tricalcium phosphate/hydroxyapatite scaffold. Oral Surg Oral Med Oral Pathol Oral Radiol Endod. 2008;106:203-9.
-Henry PJ. Tooth loss and implant replacement. Aust Dent J. 2000;45:150-72.
-Brånemark PI. Osseointegration and its experimental background. J Prosthet Dent. 1983;50:399-410.
-Sargozaie N, Moeintaghavi A, Shojaie H. Comparing the Quality of Life of Patients Requesting Dental Implants Before and After Implant. Open Dent J. 2017 31;11:485-491.
-Berber I, Erkurt MA, Kuku I, et al. An unexpected complication of bone marrow aspiration and trephine biopsy: arteriovenous fistula. Med Princ Pract. 2014;23:380-383.
-Dimitriou R, Mataliotakis GI, Angoules AG, Kanakaris NK, Giannoudis PV. Complications following autologous bone graft harvesting from the iliac crest and using the RIA: a systematic review. Injury. 2011; 42:3-15.
-Aslan H, Zilberman Y, Kandel L, Liebergall M, Oskouian RJ, Gazit D, Gazit Z. Osteogenic differentiation of noncultured immunoisolated bone marrow-derived CD105+ cells. Stem Cells. 2006;24:1728-37.
-Sterne JA, Hernán MA, Reeves BC, Savović J, Berkman ND, Viswanathan M, et al. ROBINS-I: a tool for assessing risk of bias in non-randomised studies of interventions. BMJ. 2016;355: i4919.
-Chabord P. Bone marrow mesenchymal stem cells: historical overview and concepts. Hum Gene Ther. 210; 21;1045-56.
# Why implant therapy has not been explained? We have added three articles explaining the implant therapy (line 32-36).
Osteointegrated dental implants are increasingly used to replace missing teeth in a variety of situations ranging from the missing single tooth to complete edentulism [1]. Osseointegration in clinical dentistry depends on an understanding of the healing and reparative capacities of hard and soft tissues [2]. Dental implants have a number of advantages over other therapies; high success rate, improvement in esthetics, chewing, eating or speaking [3].
-Henry PJ. Tooth loss and implant replacement. Aust Dent J. 2000;45:150-72.
-Brånemark PI. Osseointegration and its experimental background. J Prosthet Dent. 1983;50:399-410.
-Sargozaie N, Moeintaghavi A, Shojaie H. Comparing the Quality of Life of Patients Requesting Dental Implants Before and After Implant. Open Dent J. 2017 Aug 31;11:485-491.
# Why authors included the PICO in introduction? This must be included in methods section.
Thank you for the suggestion, we have changed the PICO question to the methods section (lines 92-99)
2.2 PICO question
In this work we will focus on those studies that analyze the use of mesenchymal stem cells (MSC) for the regeneration of maxillary bone defects. We set ourselves the objective of conducting a systematic review that answers the PICO question: (Population, Intervention, Comparison, Outcome): Does the use of stem cells in maxillary bone regeneration (I) improve the rate of bone formation, quality and healing time (O), compared to regenerations that only use autologous bone, biomaterial or a combination of both (C), in patients without bone availability for implant placement (P)?
# I consider that is not necessary to include the null and alternative hypothesis. This is not an original study. Thank you for the proposal, we have eliminated the null and the alternative hypothesis.
Deleted text:
In this work we will focus on those studies that analyze the use of mesenchymal stem cells for the regeneration of maxillary bone defects. For this we establish the Null Hypothesis (H0): "The use of stem cells in bone regeneration improves the quality of bone formation and accelerates the regeneration process". As an Alternative Hypothesis (H1): "The use of stem cells in bone regeneration does not improve the quality of bone formation or accelerate the regeneration process". In order to answer these hypotheses, we set ourselves the objective of conducting a systematic review that answers the PICO question: (Population, Intervention, Comparison, Outcome): Does the use of stem cells in maxillary bone regeneration (I) improve the rate of bone formation, quality and healing time (O), compared to regenerations that only use autologous bone, biomaterial or a combination of both (C), in patients without bone availability for implant placement (P)?
METHODS
# This review has not been registered in PROSPERO or similar:
This review is a recent initiative, so we have not register it yet.
# The review's phases were made by peer?
Yes, we have done it by peers. We have added it in the material and methods (page 2, lines 88-91):
Articles were selected by two blinded reviewers (JVR and SEM). Firstly, they reviewed titles and abstracts (phase‐1). If papers were considered eligible for inclusion, a full‐text reading was blindly performed by the same reviewers (phase‐2). In case of disagreements, a third and fourth reviewers (JMCS and EVO) were involved to make the final decision and agreed upon with JLL.
# Insufficient databases. WOS and CINAHL Complete not included.
After you made this contribution, we did a search in these two data bases.
In CINAHL we have found two articles that do not fit in our review.
In WOS we have found 470 articles. We read the titles and abstracts, and the full text when necessary:
- 1 article was already included in our work
- 6 were reviews of the topic
- 6 were in animal
- 5 do not use the sternum or the iliac crest to the aspirate
- 1 was a case report
- 1 fit with our inclusion/exclusion criteria so we decided to include it into our research. We were able to find that study by manual search in the pubmed/medline database. The study is:
-Shayesteh YS, Khojasteh A, Soleimani M, Alikhasi M, Khoshzaban A, Ahmadbeigi N. Sinus augmentation using human mesenchymal stem cells loaded into a beta-tricalcium phosphate/hydroxyapatite scaffold. Oral Surg Oral Med Oral Pathol Oral Radiol Endod. 2008;106:203-9.
# Why authors don't have searched in grey literature?
We have searched in grey literature, but when we found articles in other data bases that provider greater scientific rigor, we decided not to include the grey literature articles in our review.
# The search strategy proposed is very poor and non-specific. Please, improve the inclusion and exclusion criteria. Remember, exclusion criteria are not the opposite to inclusion criteria.
We have improved out inclusion/exclusion criteria (Page 3, line 100-105)
Inclusion criteria: Human Studies; No language and time restrictions; Clinical trials; Articles that present information on the bone marrow aspiration site (iliac crest or sternum), method of obtaining it, processing, handling of the cells obtained, the type of regeneration and results.
Exclusion criteria: Bibliographic reviews; Meta-analysis; About a case; Case series; Articles that did not deal with feedbacks; Animal studies; Studies that the MSC provide from de pulp/adipose tissue.
# To assess the quality with the JADAD scale is not sufficient. PRISMA review is not a quality assessment, unless a formal consideration to report the findings.
Please, add more information about the statistical analysis. Publication bias risk? WMD?? Have you considered the precision level?
To improve our review, we have added another tool to evaluate the risk of bias [Risk Of Bias In Non-randomized Studies – of Interventions (ROBINS-I) assessment tool. We have included it in the material and methods (Page 3, line 107-108):
“table 2 represents the risk of bias in non-randomized studies – of interventions (ROBINS-I) assessment tool”
And the results in Page 4, lines 131-132 and table 2:
“table 2 represents the risk of bias in non-randomized studies – of interventions (ROBINS-I) assessment tool”
You can see the new table in the document
RESULTS:
# The studies included in the review analysis must not been included before the results. Authors use the studies included to write the introduction.
We have removed the references of the studies used in our results. The references were:
- Kaigler D, Avila-Ortiz G, Travan S, Taut AD, Padial-Molina M, Rudek I, Wang F, Lanis A, Giannobile WV. Bone engineering of maxillary sinus bone deficiencies using enriched CD90+ stem cell therapy: A randomized clinical trial. J Bone Miner Res. 2015;30:1206-16.
- Ueda M, Yamada Y, Kagami H, Hibi H. Injectable Bone Applied for Ridge Augmentation and Dental Implant Placement: Human Progress Study. Implant Dent. 2008; 17:82-90.
- Wildburger A, Payer M, Jakse N, Strunk D, Etchard-Liechtenstein N, Sauerbier S. Impact of autogenous concentrated bone marrow aspirate on bone regeneration after sinus floor augmentation with a bovine bone substitute – a split-mouth pilot study. Clin. Oral Impl. Res. 2014; 25: 1175-81.
- Gjerde C, Mustafa K, Hellem S, Rojewski M, Gjengedal H, Yassin MA, Feng X, Skaale S, Berge T, Rosen A, Shi X, Ahmed AB, Gjertsen BT, Schrezenmeier H, Layrolle P. Cell therapy induced regeneration of severely atrophied mandibular bone in a clinical trial. Stem Cell Res Ther. 2018;9: 213.
- Sauerbier S, Stricker A, Kuschnierz J, Bühler F, Oshima T, Xavier SP, Schmelzeisen R, Gutwald R. In Vivo Comparison of Hard Tissue Regeneration with Human Mesenchymal Stem Cells Processed with Either the FICOLL Method or the BMAC Method. Tissue Eng Part C Methods. 2010;16:215-23.
- Bajestan MN, Rajan A, Edwards SP, Aronovich S, Cevidanes LHS, Polymeri A, Travan S, Kaigler D. Stem cell therapy for reconstruction of alveolar cleft and trauma defects in adults: A randomized controlled, clinical trial. Clin Implant Dent Relat Res. 2017;19:793-801.
# Why in the PRISMA flow chart appear six studies and, in the text, only five?
We want to apologize it was a mistake, we have done the changes. With the addition of a new study are 5 articles + 2 articles. We have corrected it in the figure 1.
You can see the new figure in the document
# Report of the results improvable.
We have changed by adding a new article and a figure to make it easier to follow
-Shayesteh YS, Khojasteh A, Soleimani M, Alikhasi M, Khoshzaban A, Ahmadbeigi N. Sinus augmentation using human mesenchymal stem cells loaded into a beta-tricalcium phosphate/hydroxyapatite scaffold. Oral Surg Oral Med Oral Pathol Oral Radiol Endod. 2008;106:203-9.
Figure 2 (page 10) you can see it in the document.
What provided a meta-analysis with two studies? And with only one study?
I would rethink a new way of writing the results, as well as assess the need for meta-analysis. A review with 5-6 + 2 studies is low in scope, but if done well it can be very interesting.
Now we have 5+3 articles. Unfortunately, we were only able to analyze 2 studies due to the heterogeneous methodology that the different studies used. We highlight it in the limitations of the review (line 341-342).
For the heterogeneity of the results, in our work, we were only able to analyze 2 studies.
REFERENCES
# Only 19 references in a review???
We have added more references to improve our review. The added references are:
- Genova T, Roato I, Carossa M, Motta C, Cavagnetto D, Mussano F. Advances on Bone Substitutes through 3D Bioprinting. Int J Mol Sci. 2020 Sep 23;21(19):7012.
-Bae MS, Kim JE, Lee JB, Heo DN, Yang DH, Kim JH, Kwon KR, Bang JB, Bae H, Kwon IK. ZrO2 surface chemically coated with hyaluronic acid hydrogel loading GDF-5 for osteogenesis in dentistry. Carbohydr Polym. 2013;92:167-75.
-Catauro M, Barrino F, Dal Poggetto G, Milazzo M, Blanco I, Ciprioti SV. Structure, drug absorption, bioactive and antibacterial properties of sol-gel SiO2/ZrO2 materials. Ceramics International, 2020; Materials, 2020, 13.2: 394.
- Soria-Juan B, Escacena N, Capilla-González V, Aguilera Y, Llanos L, Tejedo JR, et al.; Collaborative Working Group “Noma Project Team”. Cost-Effective, Safe, and Personalized Cell Therapy for Critical Limb Ischemia in Type 2 Diabetes Mellitus. Front Immunol. 2019 Jun 4;10:1151.
-Shayesteh YS, Khojasteh A, Soleimani M, Alikhasi M, Khoshzaban A, Ahmadbeigi N. Sinus augmentation using human mesenchymal stem cells loaded into a beta-tricalcium phosphate/hydroxyapatite scaffold. Oral Surg Oral Med Oral Pathol Oral Radiol Endod. 2008;106:203-9.
-Henry PJ. Tooth loss and implant replacement. Aust Dent J. 2000;45:150-72.
-Brånemark PI. Osseointegration and its experimental background. J Prosthet Dent. 1983;50:399-410.
-Sargozaie N, Moeintaghavi A, Shojaie H. Comparing the Quality of Life of Patients Requesting Dental Implants Before and After Implant. Open Dent J. 2017 31;11:485-491.
-Berber I, Erkurt MA, Kuku I, et al. An unexpected complication of bone marrow aspiration and trephine biopsy: arteriovenous fistula. Med Princ Pract. 2014;23:380-383.
-Dimitriou R, Mataliotakis GI, Angoules AG, Kanakaris NK, Giannoudis PV. Complications following autologous bone graft harvesting from the iliac crest and using the RIA: a systematic review. Injury. 2011; 42:3-15.
-Aslan H, Zilberman Y, Kandel L, Liebergall M, Oskouian RJ, Gazit D, Gazit Z. Osteogenic differentiation of noncultured immunoisolated bone marrow-derived CD105+ cells. Stem Cells. 2006;24:1728-37.
-Sterne JA, Hernán MA, Reeves BC, Savović J, Berkman ND, Viswanathan M, et al. ROBINS-I: a tool for assessing risk of bias in non-randomised studies of interventions. BMJ. 2016;355: i4919.
-Chabord P. Bone marrow mesenchymal stem cells: historical overview and concepts. Hum Gene Ther. 210; 21;1045-56.

Reviewer 4 Report
The design of study is mostly sound, and manuscript is generally well-written. However, the reviewer has comments on the manuscript.
Title
In this review, is your purpose of reviewing the clinical application or clinical efficacy? In line 241, the aim is to evaluate to efficacy of the use of stem cells.
Abstract
L15: a multitude of techniques have been developed…
I cannot find why you did your study and your study objects (by PRISMA checklist).
L21: I cannot find the number of participants and key results. The authors used forest plot to represent the results regarding new bone formation and residual content of biomaterial (Fig 2, 3). However, there is no information about the detailed results in the Abstract. Please describe it.
L24: In Results, the percentage of bone formation was not statistically significant (L227). How can you say mesenchymal stem cells in bone regenerations provides benefits in the quality of regenerated bone?
Is reviewing the different techniques for obtaining and processing these stem cells your aim?
Introduction
L33 of the treatment. -> erase .
L66 the regeneration results [1,6,7] -> [1, 6, 7]
L76: You mentioned your study focus on maxillary bone regeneration. However, among the references included (5, 11, 13, 4, 1, 12, 10), reference 10 studied mandibles. Did you confirm that all the other references for study design?
Materials and Methods
L88: In Figure 1, you excluded studies older than 15 years and duplicate articles. But I cannot find that in your exclusion criteria.
L103: Please check if writing x2, I2 fits the technical law. (or χ2 / I2)
Results
L110: You mentioned 5 articles met your inclusion criteria, but 6 publications are shown in Figure 1.
Figure 1: I cannot follow your description about “1 short implants and prosthesis refill”.
L125: Of the participants, 83 were performed using mesenchymal cells mixed with biomaterials, and 39 were done only with regeneration techniques. If it is possible to compare these two groups' clinical results in table or fibure, it would be better to explain the results to the reader.
L126: If you describe BioOss for the first time in this manuscript, please describe the detail of the manufacturer.
L229: Since I am not a dentist, there are limitations to the interpretation of the conclusions. Is residual content shown in Figure 3 clinically sound when there is more residual content? However, I cannot find an explanation for this in the front part of the Results. Also, if the control group's results were better, isn't it clinically problematic? This content is not included in the Discussion.
Table 1.
Double blind -> Please do not cut the word.
Appropiatte -> Appropriate
randomization -> Please do not cut the word.
Appropiatte -> Appropriate
puntuation -> punctuation
Table 2. Why is only Ueda's research in bold? Isn't this a format like table heading that was accidentally repeated?
Why do you describe M: Women, H: Man? Normally M if Male and F is Female.
In the study of Wildburger, what does Ilion represent for?
In the study of Gjerde, what does madre represent for?
Table 3. Write the abbreviations for BMAC and FICOLL under the table.
Discussion
L244: How is the rate of complication? Can you compare the rate with other procedures?
L250: Is the machine you use to cultivate stem cells at a price that every doctor can afford? Is it available in all countries? Please state that the cost-effectiveness may not be right.
There is a need to describe what size scaffold or stem cell is recommended for clinical use.
The authors described the higher % of CD90 cells, the better bone quality is obtained (line 162). However, they did not discuss about biomarkers as CD44, CD90, CD73 or CD105 in Discussion section.
Conclusions
L293: Isn't there insufficient evidence to describe the results as favorable?
I would encourage the authors to clarify why their findings are important, meaningful.
Author Response
The design of study is mostly sound, and manuscript is generally well-written. However, the reviewer has comments on the manuscript.
Thank you very much for your comments and suggestions.
-The authors reply:
# TITLE
In this review, is your purpose of reviewing the clinical application or clinical efficacy? In line 241, the aim is to evaluate to efficacy of the use of stem cells.
Our purpose was to evaluate the efficacy compared with other techniques, so we have changed the title to make it more accurate. The new title is:
“Clinical efficacy of mesenchymal stem cells in bone regeneration in oral implantology. Systematic review and meta-analysis”
# ABSTRACT
L15: a multitude of techniques have been developed…
I cannot find why you did your study and your study objects (by PRISMA checklist).
L15-17. We have added the aim of the review in the abstract
“The aim of this review was to evaluate the efficacy of stem cells in maxillary bone regenerations for implant rehabilitation and to review the different techniques for obtaining and processing these cells.”
L21: I cannot find the number of participants and key results. The authors used forest plot to represent the results regarding new bone formation and residual content of biomaterial (Fig 2, 3). However, there is no information about the detailed results in the Abstract. Please describe it.
L21-24. We added the number of patients and the results of the forest plots to enhance the abstract.
The results from the meta-analysis (18 patients) did not provide us with significant differences despite the percentage of bone formed in the maxillary sinus favoring the stem cell group and the analysis of the percentage of residual Bio-Oss® showed results favoring the control group.
L24: In Results, the percentage of bone formation was not statistically significant (L227). How can you say mesenchymal stem cells in bone regenerations provides benefits in the quality of regenerated bone?
L21-24. The results are not statistically significant but there is a partial improvement in the quality of regenerated bone using MSC compared with the group that do not use MSC.
Is reviewing the different techniques for obtaining and processing these stem cells your aim?
Reviewing of the techniques for obtaining and processing the stem cells is also part of our specific aim, but we explained it in the results.
# INTRODUCTION
L33 of the treatment. - erase.
Done
L66 the regeneration results [1,6,7] - [1, 6, 7]
Done
L76: You mentioned your study focus on maxillary bone regeneration. However, among the references included (5, 11, 13, 4, 1, 12, 10), reference 10 studied mandibles. Did you confirm that all the other references for study design?
When we referred to superior and inferior maxillary (page 4).
MATERIALS AND METHODS
L88: In Figure 1, you excluded studies older than 15 years and duplicate articles. But I cannot find that in your exclusion criteria.
We want to apologize for the mistake. There was an error in the figure 1. We do not make a restriction on time (page 4)
You can see the new flow chart in th document
L103: Please check if writing x2, I2 fits the technical law. (or χ2 / I2)
It is changed for χ2 and/or I2 in lines: L115, L270, L273
# Results
L110: You mentioned 5 articles met your inclusion criteria, but 6 publications are shown in Figure 1.
As mentioned before, we were sorry for the mistake in figure 1. But we have incorporated a new article. So finally if there are 6
Figure 1: I cannot follow your description about “1 short implants and prosthesis refill”.
As mentioned before, we were so sorry for the mistake that we made in figure 1. We replaced it with the correct one.
L125: Of the participants, 83 were performed using mesenchymal cells mixed with biomaterials, and 39 were done only with regeneration techniques. If it is possible to compare these two groups' clinical results in table or figure, it would be better to explain the results to the reader.
L245-247. We made a figure comparing the test group and the control group to make the information more visual.
Regarding the new bone formation, studied by 4 articles [20, 22-24], that comprise a total of 60 regenerations (42 test group and 18 control group) in which the new bone formation was 22, 94% and 13.8% for each article respectively. Figure 2.
You can see the new figure in the document
L126: If you describe BioOss for the first time in this manuscript, please describe the detail of the manufacturer.
We have added the details in line 152
[Geistlich Bio-Oss®, Inibsa Dental, Barcelona, Spain]
L229: Since I am not a dentist, there are limitations to the interpretation of the conclusions. Is residual content shown in Figure 3 clinically sound when there is more residual content? However, I cannot find an explanation for this in the front part of the Results. Also, if the control group's results were better, isn't it clinically problematic? This content is not included in the Discussion.
Line 317-318. We have attached an explanation in the discussion.
Finding a high amount of residual biomaterial leads us to find a lower quality of overall bone since there will be a subsequent lower quantity of neoformed native vital bone.
# Table 1.
Double blind - Please do not cut the word.
Appropiatte - Appropriate
Randomization - Please do not cut the word.
Appropiatte - Appropriate
Puntuation - punctuation
All changes are done in Table 1.
# Table 2. - with the changes, now is table 3.
Why is only Ueda's research in bold? Isn't this a format like table heading that was accidentally repeated?
It was a mistake; we have changed it (page 7).
Why do you describe M: Women, H: Man? Normally M if Male and F is Female.
It is changed in table 3 (page 7)
In the study of Wildburger, what does Ilion represent for?
We have changed for Ilium (page 7)
In the study of Gjerde, what does madre represent for?
We have erased it madre
# Table 3. - with the changes now is table 4
Write the abbreviations for BMAC and FICOLL under the table.
We have written above the tables.
BMAC: Bone Marrow Aspirate Concentrate; FICOLL: bone marrow-derived mononuclear cells isolation by synthetic polysacaryd
# DISCUSSION
L244: How is the rate of complication? Can you compare the rate with other procedures?
We have compared the rate of complications of bone marrow aspirate amongst techniques for obtaining autologous bone grafts. (line 287-289)
This technique presents a low rate of complications (0,05%) and morbidity for the patient [25]; which gives it a clear advantage over techniques for obtaining autologous bone [6,20], which frequently involves minor complications (9-39%) and occasionally severe complications (0.76-25%) [26].
We attached these two references.
- Berber I, Erkurt MA, Kuku I, et al. An unexpected complication of bone marrow aspiration and trephine biopsy: arteriovenous fistula. Med Princ Pract. 2014;23:380-383.
- Dimitriou R, Mataliotakis GI, Angoules AG, Kanakaris NK, Giannoudis PV. Complications following autologous bone graft harvesting from the iliac crest and using the RIA: a systematic review. Injury. 2011; 42:3-15.
L250: Is the machine you use to cultivate stem cells at a price that every doctor can afford? Is it available in all countries? Please state that the cost-effectiveness may not be right.
In lines 331-336, we have made a comment about this topic based on this article, that we have added in the bibliography:
-Soria-Juan B, Escacena N, Capilla-González V, Aguilera Y, Llanos L, Tejedo JR, et al.; Collaborative Working Group “Noma Project Team”. Cost-Effective, Safe, and Personalized Cell Therapy for Critical Limb Ischemia in Type 2 Diabetes Mellitus. Front Immunol. 2019 Jun 4;10:1151. Erratum in: Front Immunol. 2020 Sep 02;11:2029.
There is a need to describe what size scaffold or stem cell is recommended for clinical use.
In lines 200-213 there is an explanation about concentrations and quantities.
The concentrations of the different biomaterials vary according to the study: Three studies [18, 22, 23] use BioOss®; 2g of BioOss® (0.25-1mm, particle size) with 3mL of mononuclear cells (40x106 cells / mL specified in study [22]) with 1mL of human thrombin [22, 23], and the same combination but without specifying quantities [18]. In two studies [17, 21] β-TCP from Cerasorb® was used; β-TCP in 1: 1 volume ratio with mononuclear cells, β-TCP in 1: 1 volume ratio with mononuclear cells, a total of approximately 15-80x106 cells per graft (5-10x106 cells / mL) [17]; and 2-5 mL of β-TCP with 15-44x106 cells / mL of mononuclear cells [21]. In the study by Ueda et al. [18] a combination of 3.5mL platelet-rich plasma (PRP) with 1x107 mononuclear cells / mL mixed with 500μL of calcium chloride and thrombin, for a total of 1.5-5.8gr per graft. Gjerde et al. [19] combined 5mL of BCP (MBCP®) consisting of 20% hydroxyapatite and 80% β-TCP (in 0.5-1mm granules) with 100x106 mononuclear cells (20x106 cells / ml). In the study by Sauerbier et al. [20] a combination of BioOss® (0.25-1mm) with thrombin-enriched mononuclear cells was used, without specifying the number of cells per mL. Finally, Shayetesh et al. [24] used cubes composed by β-TCP/hydroxyapatite, as scaffold, mixed with 0.2ml (5x105 cells) of MSC.
The authors described the higher % of CD90 cells, the better bone quality is obtained (line 162). However, they did not discuss about biomarkers as CD44, CD90, CD73 or CD105 in Discussion section.
L303-304 We have added information in discussion about another biomarkers.
Positive expression of biomarkers CD105, CD73 and CD90 are involve in osteogenic process by promoting bone formation [31].
# CONCLUSIONS
L293: Isn't there insufficient evidence to describe the results as favorable?
I would encourage the authors to clarify why their findings are important, meaningful.
We have changed the conclusions to clarify the results (line 346-354).
In conclusion, despite the limitations of the present study, bone regeneration through the application of stem cells in surgery and oral implantology show favorable results, although not always statistically significant. This is an innovative technique with small number and low quality of trials, lack of controls and short follow-up. Instead of these limitations, this technique seems that could bring us an alternative to autologous bone grafts because of this low rate of morbility and complications, reduction of the healing time; and its capacity to form a high-quality bone. In future, with more randomized clinical trials should be carried out, establishing homogeneous treatment protocols, with larger samples and longer follow-up periods could show us the real potencial of MSC to develop a minimally invasive treatment to favor high quality bone tissue regeneration.

Reviewer 5 Report
I can not recommend this manuscript for further evaluation to International Journal of Environmental Research and Public Health. The main problem is a very few number of articles that met Authors` inclusion criteria (only 5). Thus in my opinion this review manuscript is not sufficient for this Journal.
Author Response
I cannot recommend this manuscript for further evaluation to International Journal of Environmental Research and Public Health. The main problem is a very few number of articles that met Authors` inclusion criteria (only 5). Thus in my opinion this review manuscript is not sufficient for this Journal.
We are very grateful that you have reviewed our work. We are aware that there are few references, but it is a field in which there is little research on the matter. We wanted to put restrictive inclusion and exclusion criteria to homogenize as much as possible, and achieve more consistent results. Revising the subject again we have found another article that fits our criteria and we have added to our analysis; and have also attached more articles to the review.
We would appreciate that with this new incorporation and all the changes we made, you could reconsider the possibility of being able to publish in Int. J. Environ. Res. Public Health.
Round 2
Reviewer 1 Report
I am satisfied with authors' changes to the text. I believe that the article is now suitable for publication in ijerph. The introduction provides sufficient details to properly describe the context of the research. Methods are well organized and results are clearly described. Discussion and conclusion are supported by results and take into consideration all relevant literature
Author Response
I am satisfied with authors' changes to the text. I believe that the article is now suitable for publication in ijerph. The introduction provides sufficient details to properly describe the context of the research. Methods are well organized and results are clearly described. Discussion and conclusion are supported by results and take into consideration all relevant literature.
Thank you very much for accepting our manuscript to be published in ijerph.

Reviewer 3 Report
Thank you for your answer. It is very difficult to read the manuscript with the changes control.
The aim in the abstract is not only to evaluate the effect, unless to collect the available evidence.
The results from the meta-analysis must include the number of studies and the pooled effect, not only the patients. Only 18?
Why authors exclude PICO in a different subsection of search strategy? Why authors have not included the study design in PICOS?
About PROSPERO, I consider that it must be registered previously. Although it was a recent initiative.
How can I sure that really it has been made by peer? The name of the authors that have participated in each stage does not appear.
It is necessary to add WOS and CINAHL
The inclusion criteria are not correct. You add clinical trials but later use a methodological quality scale for not interventions study…
Please, use PEDRO or Cochrane Risk of bias tool.
You have added references without explain why.
I consider that this work must be more improved.
Author Response
# Thank you for your answer. It is very difficult to read the manuscript with the changes control.
Thank you for revising our manuscript. We apologize and we send you another manuscript without change control and the modifications in red.
# The aim in the abstract is not only to evaluate the effect, unless to collect the available evidence.
We do not understand your appreciation; the aim of the review is to evaluate the effect of stem cells collecting the available evidence that we could find in the current literature.
# The results from the meta-analysis must include the number of studies and the pooled effect, not only the patients. Only 18?
Unfortunately the data obtained are not excessive but the published studies are limited.
Line 273 and 278: The results of the meta-analysis include 25 patients (36 regenerations) of two studies.
# Why authors exclude PICO in a different subsection of search strategy? Why authors have not included the study design in PICOS?
We have included the PICO in the section of search strategy as you recommended (Lines 95-101)
In this work, we will focus on those studies that analyze the use of mesenchymal stem cells (MSC) for the regeneration of maxillary bone defects. We set ourselves the objective of conducting a systematic review that answers the PICO question: (Population, Intervention, Comparison, Outcome): Does the use of stem cells in maxillary bone regeneration (I) improve the rate of bone formation, quality, and healing time (O), compared to regenerations that only use autologous bone, biomaterial or a combination of both (C), in patients without bone availability for implant placement (P)?
# About PROSPERO, I consider that it must be registered previously. Although it was a recent initiative.
As we mentioned in the previous answer, we apologize for not having entered our review in the PROSPERO database. A registry system that, although it has been working regularly since 2011, [(http://www.crd.york.ac.uk/prospero/)),and for different reasons, has not reached enough popularity among researchers. We have followed the PRISMA guidelines, 2014 [(http://www.crd.york.ac.uk/prospero/)] and we have registered our review in PROSPERO system, but still, we have not gotten answers. We hope and appreciate your understanding in this regard.
# How can I sure that really it has been made by peer? The name of the authors that have participated in each stage does not appear.
Lines 88-93: We have added the names of the authors that have participated in each stage.
Articles were selected by two blinded reviewers (Valls-Roca-Umbert J, –JVR-) and Egido-Moreno S, –SEM-)). Firstly, they reviewed titles and abstracts (phase‐1). If papers were considered eligible for inclusion, a full‐text reading was blindly performed by the same reviewers (phase‐2). In case of disagreements, a third and fourth reviewers (Céspedes-Sánchez JM, –JMCS-) and Velasco-Ortega E, -EVO-) were involved to make the final decision and agreed upon with López-López J -JLL-.
# It is necessary to add WOS and CINAHL
We have added WOS and CINAHL databases to our research (lines 79-84)
A bibliographic review is carried out in the Pubmed / Medline (NCBI), Cochrane, Scielo, Web of Science (WOS), Cumulative Index to Nursing and Allied Health Literature (CINAHL) and Scopus databases of articles without applying restriction on the publication date, performed until October 31, 2020 that contains information of the use of stem cells obtained from the bone marrow for bone regeneration in the jaws. The following Mesh terms were used combined by the Boolean operator "AND": “dental implants” AND “stem cells” AND “bioengineering”.
We have modificated the results (lines 123-130) and the figure 1 (flow chart) with the addition of these data bases.
Through our search strategy in Medline/Pubmed, Scopus, Scielo, WOS, CINAHL, and Cochrane 382 articles were obtained. After eliminating the animal studies, 294 publications remained. After reading the titles and abstracts, 240 papers were discarded.
Of these, we eliminated 4 case series, 8 case reports, 16 reviews, and 18 studies that obtained stem cells from dental pulp or adipose tissue. Finally, after eliminating the duplicated articles, there were 6 articles [17-22] that met our inclusion criteria. Two additional articles resulting from the manual search were included for the interest of the review (Rickert D et al. [23] and Wildburger A et al. [24]) (Fig. 1).
#The inclusion criteria are not correct. You add clinical trials but later use a methodological quality scale for not interventions study…
Please, use PEDRO or Cochrane Risk of bias tool.
In the inclusion criteria we add clinical trials and we analyze the risk of bias with JADAD scale and Robins Tool I (Cochrane method) (table 2) both directed to analyze clinical trials.
# You have added references without explain why.
Other reviewers requested us to add some references to improve some previous points; aspects that we answered in previous review that the reviewers have kindly accepted.
# I consider that this work must be more improved.
We would appreciate if you could review the manuscript again and reconsider the possibility of being able to publish in Int. J. Environ. Res. Public Health. Very thankful

Reviewer 5 Report
The manuscript was strongly corrected, and now this manuscript is acceptable.
Minor comments:
Please check the manuscript carefully, it is necessary to exclude the misspellings.
Author Response
The manuscript was strongly corrected, and now this manuscript is acceptable.
Minor comments:
Please check the manuscript carefully, it is necessary to exclude the misspellings.
Thank you very much for reconsidering your decision.
We have reviewed the manuscript with a native colleague to improve the English.
